# Are Sexual Assaults Related to Functional Somatic Disorders? A Cross-Sectional Study

**DOI:** 10.3390/ijerph20206947

**Published:** 2023-10-20

**Authors:** Sofie Abildgaard Jacobsen, Lisbeth Frostholm, Cæcilie Böck Buhmann, Marie Weinreich Petersen, Eva Ørnbøl, Thomas Meinertz Dantoft, Anne Ahrendt Bjerregaard, Lene Falgaard Eplov, Tina Birgitte Wisbech Carstensen

**Affiliations:** 1Research Clinic for Functional Disorders and Psychosomatics, Aarhus University Hospital, 8200 Aarhus N, Denmark; lisbeth.frostholm@aarhus.rm.dk (L.F.); mawept@rm.dk (M.W.P.); eva.oernboel@aarhus.rm.dk (E.Ø.); tinacars@rm.dk (T.B.W.C.); 2Department of Clinical Medicine, Aarhus University, 8000 Aarhus C, Denmark; 3Mental Health Services, Capital Region of Denmark, 1172 Copenhagen, Denmark; caecilie@traumeklinikken.dk; 4Center for Clinical Research and Prevention, Bispebjerg and Frederiksberg Hospital, Capital Region of Denmark, 2000 Frederiksberg, Denmark; thomas.meinertz.dantoft@regionh.dk (T.M.D.); anne.ahrendt.bjerregaard@regionh.dk (A.A.B.); 5Department of Epidemiology Research, Statens Serum Institute, 2300 Copenhagen, Denmark; 6Copenhagen Research Centre for Mental Health—CORE (Eplov), Mental Health Centre Copenhagen, Copenhagen University Hospital, 2100 Copenhagen, Denmark; lene.falgaard.eplov@regionh.dk

**Keywords:** sexual assault, functional somatic disorder, functional somatic syndromes, cross-sectional study, somatic symptoms, emotional distress

## Abstract

An increasing number of sexual assaults (SAs) are being reported. This study investigated associations between SA and FSD, conceptualized as bodily distress syndrome (BDS), and five functional somatic syndromes (FSSs): chronic widespread pain (CWP), irritable bowel (IB), chronic fatigue (CF), multiple chemical sensitivity (MCS), and whiplash-associated disorder (WAD). Participants (*n* = 7493) from the population-based cohort Danish Study of Functional Disorders (DanFunD) completed questionnaires on FSD, emotional distress, SA, and sociodemographics. Risk ratios (RRs) for each FSD and emotional distress were calculated in nine models with SA as the primary exposure using generalized linear models with binomial family and log link and were adjusted for other potential risk factors. The results showed that SA was associated with single-organ FSD (RR = 1.51; 95% CI = 1.22–1.87), multi-organ FSD (RR = 3.51; 95% CI = 1.89–6.49), CWP (RR = 1.28; 95% CI = 0.83–1.98), IB (RR = 2.00; 95% CI = 1.30–3.07), CF (RR = 1.81; 95% CI = 1.42–2.32), WAD (RR = 2.62; 95% CI = 1.37–5.03), MCS (RR = 3.04; 95% CI = 1.79–5.17), emotional distress (RR = 1.75; 95% CI = 1.21–2.54), and health anxiety (RR = 1.65; 95% CI = 1.10–2.46). Overall, SA victims experienced significantly more somatic symptoms than individuals not exposed to SA. Adjusting for physical and emotional abuse did not change the observed associations. Our results suggest a large impact of SA on the overall somatic and mental health of SA victims. Due to the cross-sectional study design, further studies are required.

## 1. Introduction

In recent years, an increasing number of victims of sexual assault (SA) [1,2] and rape [3] have been reported in Denmark. SA presents a public health concern with WHO estimating a lifelong prevalence of 23% for women in Northern Europe experiencing intimate partner violence with physical or sexual assault, and an estimated 10% of Europeans experiencing non-partnered sexual assault in their lifetime [4]. Previous studies have found that women exposed to SA have a higher risk of experiencing chronic pain in several body regions [5,6]. A few prospective studies have investigated SA victims and found an increase in severe acute pain conditions in the whole body a week after the assault [7,8] and up to 3 months after [6]. Furthermore, it was found that approx. 40% (*n* = 30) experienced pain from more than one body region [6]. 

Functional somatic disorders (FSDs) are characterized by a heavy symptom load and disability. The etiology of FSD often includes a range of biopsychosocial risk factors [9], ranging from somatic illness [10], physical trauma, such as head injuries [11] or neck pain caused by traffic collisions [12], maladaptive illness behavior, negative illness belief [13], childhood adversity [14] and SA [15]. FSDs associated with SA include pain and gastrointestinal syndromes, fibromyalgia, also known as chronic widespread pain (CWP), and irritable bowel syndrome (IB) [16,17,18,19,20,21,22], whereas symptoms such as headache and fatigue seem to be rare [5,15].

Furthermore, co-occurrence of mental disorders is known to affect the outcome of symptoms in FSDs. In a review, Mason and Rodrick reported associations between SA and anxiety, depression, PTSD, and psychosomatic complaints [23], while other studies found similar associations between SA and health anxiety [24,25]. However, a prospective study by Ulirsh et al. found that correlations between PTSD and somatic symptoms in victims were only low to moderate, suggesting separate associations and no interaction between PTSD and pain [6].

Other than SA in childhood, adverse childhood conditions may be potential confounders of the association between SA and FSD [26,27,28,29], and physical and emotional abuse has been shown to influence the association as well [30], with several studies showing a high prevalence of physical abuse history in SA victims [16,17,19,27,31]. Additionally, studies suggest that SA in childhood does not have stronger associations than SA in adolescence [5,32].

Furthermore, there are strong associations between SA severity and symptom severity, as victims of SA seem prone to experience a high frequency of symptoms and a high pain severity [17,20,33], e.g., one study showed that fibromyalgia was only associated with SA in victims of rape and not in victims of minor assaults [5]. 

We do not know how the association between SA and FSD has developed; however, there are probably numerous psychological and psychosomatic explanations for the mechanisms, such as deficits in emotion regulation [34] and repression as seen in psychodynamic psychotherapy such as ISTDP (Intensive Short Term Dynamic Psychotherapy), and EAET (Emotional Awareness and Expression Therapy) [35,36]. Other theories suggest the possibility of sensitization of pain or other somatic symptoms after sexual trauma [37]. Furthermore, it is believed that social vulnerability, affected by events such as SA, produces toxic stress, which has been observed to promote heritable changes within the epigenome system [38,39] and has also been linked to FSDs.

The association between SA and FSD is poorly investigated as pointed out in some reviews [40] and meta-analyses [22,30]. In addition, results are inconsistent as studies include small populations and use different measures of either SA, FSD [30,40], or somatic symptoms [40]. This complicates comparability.

Therefore, there is a need to investigate the association between SA and FSDs in large, unselected population cohort studies. The current study includes data from the Danish Study of Functional Disorders (DanFunD), which is the first large coordinated epidemiological population-based study with an exclusive focus on FSDs. The aim of this study is to investigate from which symptom clusters victims of sexual assault experience functional symptoms. Moreover, we hypothesized that the severity of the assault is associated with the number and severity of functional symptoms, as rape might result in more impairment than, e.g., groping. As suggested by Ulirsh et al., we hypothesized that mental health has an effect on the association and included a minor investigation on poor mental health data.

To solve the inconsistency of delimitations of the various FSDs in previous studies, we conducted this study by investigating the following hypotheses:

Primary:Being exposed to SA is associated with experiencing FSD.

Secondary:ii.SA victims have a higher prevalence and more severe single somatic symptoms than individuals not exposed to SA.iii.There is a correlation between SA severity and the development of FSD.iv.Associations between SA and FSDs are specific and distinct from associations between SA and mental health.

## 2. Materials and Methods

### 2.1. Study Design and Population

This project is part of the Danish study of Functional Disorders (DanFunD) and includes data from the DanFunD Part Two baseline cohort [41], which is a longitudinal cohort study of the epidemiology of functional somatic syndromes. A total of 25,368 people were invited, of which 7493 agreed to participate. Data were collected during 2012–2015. Participation is described in the Appendix A. The population sample is from the western part of greater Copenhagen and consists of women and men aged 18–72 years. Exclusion criteria were the following: not born in Denmark, not a Danish citizen, and pregnancy.

#### Ethics

Written informed consent was obtained from each participant before participation, and the study was approved by the Ethical Committee of Copenhagen County (Ethics Committee: H-3-2012-0015) and the Danish Data Protection Agency [41].

### 2.2. Primary Variables: Exposure

All variables were obtained from self-reported questionnaires.

#### Sexual Assault

SA was measured with two items from the Cumulative Lifetime Adversity Measure (CLAM) [42,43,44]. The variable groping was worded as: “Have you had someone touch or feel private areas of your body or touched/felt another’s private areas under force or threat” and the variable rape as: “Have you had sexual relations under force or threat”. For the linear models, we pooled the two variables and used the term “sexual assault” with one answer in either groped or rape being considered as sexual assault; furthermore, we made a categorized variable for the sensitivity analysis with groping, rape, and both.

### 2.3. Primary Variables: Outcomes

#### 2.3.1. Functional Somatic Disorder Conceptualized as Bodily Distress Syndrome (BDS)

Former studies have conceptualized FSD as both single- and multi-organ FSD (symptom-specific BDS) and medical-specialty-specific syndromes (FSSs) [45,46]. In the present study, we use the term FSD as an overall term when addressing FSD conceptualized as both BDS and FSS. We will use the term FSD when addressing FSD conceptualized only as BDS and the term FSS when addressing FSS conceptualized only as IB, CWP, CF, WAD, or MCS (see Appendix A for an overview of abbreviations). Previous research has repeatedly suggested that the delimitations of the various FSDs are inconsistent [47]. It has been argued that different approaches to defining FSDs are preferably used within epidemiological research in order to capture the diverse nature of the conditions [45]. 

FSD was measured with the 25-item Bodily Distress checklist [48,49], which is a self-report measure to assess symptoms from four clusters (cardiopulmonary, gastrointestinal, musculoskeletal, and general/fatigue symptoms) experienced during the last 12 months.

#### 2.3.2. Single-Organ FSD

If participants experienced four or more symptoms from one or two symptom clusters, they were included as a case of single-organ FSD [49,50,51].

#### 2.3.3. Multi-Organ FSD

If participants experienced four or more symptoms from at least three symptom clusters, they were included as a case of multi-organ FSD [49,50,51]. 

#### 2.3.4. FSS

The five FSS rely on data initially collected in the DanFunD study [41]; the rationale of the five specific syndromes is to include common as well as rare syndromes in the population. The five FSS all rely on validated measures often used in population-based studies, which rely on case descriptions from Petersen et al. [46], where all symptoms are assessed by questionnaires measuring symptoms experienced during the last 12 months. CWP was defined according to the criteria of the American College of Rheumatology (ACR) [52] and by White et al. [53]; IB according to the criteria of Kay and Joergensen [54]; CF according to the criteria of Chalder et al. [55]; WAD according to the criteria of Kasch et al. [56]; and MCS according to the 1999 consensus criteria with modifications by Lacour et al. [57,58].

#### 2.3.5. Somatic Symptoms

The number of somatic symptoms was measured using the 25-item Bodily Distress checklist [48,49]. When measuring prevalence, a yes/no variable was combined. Participants were categorized as not having symptoms when replying “not at all bothered” and for having symptoms when replying “a bit”, “somewhat”, quite a bit”, and “a lot”.

#### 2.3.6. Emotional Distress

Emotional distress was measured using the validated SCL-8 [59], which is a joint measure of symptoms of anxiety and depression derived from the revised 90-item Symptom Checklist (SCL-90-R) [60,61]. However, a validated cut point has not been published for this scale. Therefore, we decided on a cut point at the 95th percentile, which made a comparable case size to the FSD cases (N:357). Our 95th percentile cut indicated an 80% chance of a clinical diagnosis of anxiety or depression [62].

#### 2.3.7. Health Anxiety

Health anxiety was measured using the Whiteley-6R measurement of illness worry [63]. We decided to use a similar cut point for emotional distress at the 95th percentile to make a comparable size to the FSD cases and emotional distress. Our 95th percentile indicates high levels of health anxiety.

### 2.4. Secondary Variables: Descriptive Data and Confounders

#### 2.4.1. Age at Time of Inclusion

Measured at the time of filling out the questionnaire.

#### 2.4.2. Age at Time of SA

The variable was categorized as preschool children (0–5 years old), school children (6–11 years old), teenagers (12–18 years old), and adults (>18 years old).

#### 2.4.3. Sex

Categorized as male or female.

#### 2.4.4. Vocational Training

Vocational training was measured as years of education after elementary school by Petersen et al. [46]. The variable was obtained from the question: *“Do you have vocational training after elementary school?”*, with a yes/no answer option. The variable was divided into four groups (“no education”, “<3 years”, “3–4 years”, and “>4 years”).

#### 2.4.5. Self-Reported Social Status

Self-reported social status was measured using a 10-point rating scale. The measure was given with the following instruction: “Think of this ladder as representing where people stand in our society. At the top of the ladder are the people who are the best off—those who have the most money, most education, and best jobs. At the bottom are the people who are the worst off—who have the least money, the poorest jobs, or no jobs. The higher up you are on this ladder, the closer you are to the people at the very top and the lower you are, the closer you are to the people at the very bottom. Please mark on the rung of the ladder where you would place yourself.” [64].

#### 2.4.6. Adverse Childhood Environment

To measure negative childhood environment, which might be correlated with our primary exposure variable, we combined five questions from the CLAM questionnaire: Have you: *“Lost someone close to you to homicide?”; “Experienced serious financial difficulties (i.e., no money, food, or shelter)?”; ”Lived in dangerous housing or neighborhood?”; “Witnessed violence between your parents as a child?”; and “Experienced a tragedy or disaster in your community caused by people (a shooting, bombing, etc.)?”* An event was calculated if the participant had experienced any of the five events before the age of 18 years. Cases of “adverse childhood environment” counted if participants had at least one negative lifetime experience. Participants were categorized as yes, in a dichotomized 0/1 variable, if they had experienced any one of the events.

#### 2.4.7. Abuse

The variable abuse contains elements of both physical and emotional abuse and is a lifelong, retrospective measure. Physical abuse was measured with three items from the CLAM [42,43,44]: “*Were physically attacked or insulted; Being physically harmed as a child (hit hard enough to leave a bruise or mark, kicked, burned, etc.); Been hit or pushed by your partner/spouse*” with one answer in either item being considered as physical abuse. Emotional abuse was measured with one item from the CLAM [42,43,44]: *“Been shamed, embarrassed, or told repeatedly that you are “no good”.* We combined the variables in the analysis as a dichotomized variable, with a “yes” in either physical or emotional abuse being considered a case in the variable “Abuse”.

### 2.5. Statistical Analysis

#### 2.5.1. General Statistics

Descriptive statistics were used to display the main characteristics of the study sample and to investigate the presentation of somatic symptoms. Comparisons between groups were performed by means of $\chi^{2}$ test. Missing data on SA cases were analyzed with descriptive statistics. Stata version 16 was used for all analyses [65].

#### 2.5.2. Regression Model

Separate analyses were performed with FSD cases, including single-organ and multi-organ FSD, IB, CF, WAD, CWP, and MCS as well as a high level of emotional distress and health anxiety as dependent variables. Risk ratios (RR) for FSD cases, emotional distress, and health anxiety were estimated in nine models with sexual assault as the primary exposure using generalized linear models with binomial family and log link. 

To avoid overfitting, 10–15 cases for each explanatory parameter were estimated [66,67]. An a priori based list of potential explanatory variables was made by investigating and listing variables employed in similar studies and comparing them with what was possible in the data pool. Due to the case sizes, we decided not to include age at exposure as an explanatory variable in the adjusted analyses. 

In the final analysis, we adjusted for age at examination, self-rated social status, sex, vocational training, and adverse childhood environment. Age at examination and social status were modeled through a restricted cubic spline with 4 knots. Regarding the outcomes of emotional distress and health anxiety, we investigated if there was a similar RR for SA as to FSD in crude and adjusted analyses. We further analyzed whether emotional distress and health anxiety moderated the association between FSD and SA in crude analyses only. Finally, we performed a sensitivity analysis on the SA variable, categorizing the variable into groping, rape, and experiences of both in crude analysis only. 

In case of problems with the convergence of the above models, a generalized linear model with the Poisson family and log link was estimated instead. 

The results are presented as crude and adjusted risk ratios (RRs) with 95% CI. 

## 3. Results

### 3.1. Descriptive Analysis

The population consisted of 7493 participants, 53.9% female, and a median age of 54 (SD:13.32) years; the majority (*n* = 3202, 42.7%) of the participants reported having 3–4 years of vocational training and a median self-rated social status of 7 (Table 1). The population included 235 victims of SA consisting of 139 individuals exposed to rape and 179 individuals exposed to groping, thus with an overlap of 83 individuals between the groups. Approximately 9 in 10 cases of SA victims were female, of which 1 in 10 cases had experienced an adverse childhood environment. Half of the victims of SA had also experienced physical abuse, while approx. one third had also experienced emotional abuse, with 60.4% having experienced either physical and/or emotional abuse. Furthermore, participants experiencing SA had lower vocational training, lower self-reported social ratings, and were younger than individuals not having experienced SA. Additionally, victims of SA also experienced current emotional distress and health anxiety with 11%. Half of the victims of groping had been assaulted before the age of 12. Victims of rape were more likely to have been assaulted in adolescence and adulthood than in childhood. 

Distribution of SA victims fulfilling the criteria for FSD by syndrome: multi-organ FSD (15.5%), single-organ FSD (6%), WAD (4.7%), MCS (7.2%), CWP (8.5%), IB (9.4%), and CF (22.6%). When dividing each of the five FSSs into age groups at exposure, the trend showed the least exposure at <5 years of age and the highest at 12–18 years.

When dividing the results between sexes, we found some differences between women and men. Female victims of sexual assault were younger at baseline and experienced SA later in life into adulthood. They experienced more emotional distress and were twice as likely to experience emotional abuse than men. Male victims of sexual abuse mostly experienced SA in childhood. Male victims of rape were 10 years older at baseline than male victims of groping. Male victims were more likely to have no education and had twice as many adverse childhood environment experiences compared with females. Female and male victims of SA were similar in terms of physical abuse, self-reported social status, and health anxiety. The results for women can be found in Appendix A; the results for men are not shown due to <5 men in some of the cells.

#### Analysis of Missing SA Victims

The analysis of participants who had not completed the two items on SA (Appendix A) showed similar results to those of the total population (Table 1) on sociodemographics and outcome measures.

### 3.2. Crude and Adjusted Analysis of FSD

Regardless of delimitation, all crude analyses showed strong associations between having been exposed to SA and FSD (Table 2). When adjusting for potential confounding variables: age at the time of inclusion, sex, vocational training, social status, and an adverse childhood environment, the association between SA and CWP was attenuated. The highest risk was found in participants with multi-organ FSD (RR = 3.51; 95% CI = 1.89–6.49, *p* < 0.001) followed by MCS (RR = 3.04; 95% CI = 1.79–5.17, *p* < 0.001) (Table 2).

Dividing the sample into sexes did not alter the main results, as the adjusted regression for women was similar to that of the total population (only 26 men had experienced SA). For each measure of FSD, we found that the CI for women was very similar to those of men in the crude regression analyses. Results not shown.

### 3.3. Descriptive Analysis of Somatic Symptoms

SA victims had a significantly higher prevalence across all categories of somatic symptoms than the participants with no experience of SA (Figure 1). Viewing the symptom subgroups (general/fatigue symptoms, gastrointestinal symptoms, musculoskeletal symptoms, and cardiopulmonary symptoms), it was evident that SA victims had experienced more severe symptoms than those who had not been exposed to SA, e.g., 2.7% not exposed to SA had been “bothered a lot” by excessive fatigue compared with 7.2% of SA victims. All results were similar and statistically significant (Appendix A).

### 3.4. Analysis of Subdivisions: Rape and Groping

Overall, we found more significantly increased RRs for the association between rape and FSD than for groping. All crude results showed a significant association with all FSDs (Table 3). When adjusting for potential confounding variables: age at the time of inclusion, sex, vocational training, social status, and adverse childhood environment, the association between groping and all outcomes but multi-organ FSD and CF became non-significant. For rape, the association remained significant except in cases of CWP and WAD. There was a five times higher RR of multi-organ FSD in victims of rape (RR = 5.18; 95% CI = 2.73–9.86, *p* < 0.001) than in participants not exposed to rape. For groping, the risk for multi-organ FSD was RR = 3.76; 95% CI = 1.99–7.09, *p* < 0.001.

When examining the subdivisions of SA according to the analysis of each somatic symptom, victims of rape felt even more bothered by each symptom than victims of groping, suggesting an association between SA severity and severity of somatic symptoms (Appendix A).

We divided the variable into three groups: groping (40.9%, *n* = 96), rape (23.8%, *n* = 56), and both combined (35%, *n* = 83). Crude analysis showed the highest RRs for experiencing both groping and rape for multi-organ-FSD and CF. Furthermore, it is clear from the sensitivity analysis that rape has a higher RR compared with groping, as shown in Appendix A**.**

### 3.5. Analysis of the Mental Health Components 

SA victims had an almost 50% increased risk of experiencing emotional distress compared with participants not exposed to SA in the adjusted analysis (RR = 1.75; 95% CI = 1.21–2.54, *p* = 0.003), with a similar risk for experiencing health anxiety (RR = 1.65; 95% CI = 1.10–2.46, *p* = 0.014). The results for both components are comparable to results in FSD cases and are adjusted for the same possible confounders. 

Emotional distress (RR = 0.34; 95% CI = 0.19–0.61, *p* > 0.0001) and health anxiety (RR = 0.45; 95% CI = 0.28–0.71, *p* = 0.001) seemed to have a dampening effect on the crude interaction for single-organ FSD and SA. Similar results were seen for the interaction analysis between CF and SA with a dampening effect for emotional distress (RR = 0.39; 95% CI = 0.24–0.63; *p* < 0.0001) and for health anxiety (RR = 0.57; 95% CI = 0.36–0.92, *p* = 0.020). We found no significant interaction between FSS cases and emotional distress or health anxiety for CWP, IB, WAD, and MCS. For multi-organ FSD we found emotional distress to have a dampening effect (RR = 0.25; 95% CI = 0.08–0.75, *p* = 0.014), while there was no significant interaction between multi-organ FSD and health anxiety. 

### 3.6. Analysis of Physical and Emotional Abuse

The results of SA on various FSDs when physical and emotional abuse are included as confounding variables are shown in Table 4. Overall, the results are somewhat similar to the main analyses for SA. The tendency of increased RRs for FSD is still significant or borderline significant in the adjusted analysis for all outcomes in rape victims except for CWP and WAD.

## 4. Discussion

In this large population-based study, we found a strong association between FSD and SA. The association was true for all FSD measures, especially for multi-organ FSD and the variety of somatic symptoms. Furthermore, we found similar results for the association between SA and emotional distress. These findings implicate that SA may be a risk factor for experiencing FSD and a variety of somatic symptoms as well as emotional distress and health anxiety. Our results point towards a negative impact of SA on the overall somatic and psychological health of victims.

We did not find a significant association between CWP and SA in the adjusted analysis, which is contrary to findings from previous studies [16,17,18,19,20,21,22]. However, our CI included a possibility of a double risk. Altogether, with the critique of the various measures and study designs [22,30], previous studies do not necessarily contradict our results but merely point out the necessity to investigate the field more thoroughly. 

As the high RR for multi-organ FSD (RR = 3.51) has a wide CI, this may be due to the case sample of 84 individuals, while a more precise risk assessment may range from a double risk to a six times higher risk. This concern applies to the high RRs for MCS and WAD as only 11 and 17 individuals, respectively, with this syndrome were exposed to SA. Additionally, the CIs were wide, and the risk ratios for WAD and MCS should, therefore, be interpreted with caution. To the best of our knowledge, there is no available literature on associations between SA and WAD or MCS.

The literature on CF and SA is very sparse as only two case-control studies have been performed, both by Taylor and Jayson [28,31]. They found significant results for clinical diagnoses of CF, but none for self-reported data [31]. The association between IB and SA is the most investigated of the five syndromes included in our study, and our results are similar to those found in the literature [16,17,18,19,20].

The results for the symptomology of SA showed a higher prevalence and more severe somatic symptoms across all organ systems, indicating that sexual trauma is not solely a risk factor for somatic symptoms centered in the pelvic or gastrointestinal area of the body, but also emerges from and affects the whole body. Therefore, there is reason to believe that other syndromes than those included in this study may also be associated with SA. SA may be a general risk factor across numerous FSDs and similar diagnoses, such as somatic symptom disorder (SSD), as we found that approx. one in ten victims of SA experienced high levels of health anxiety, which is an inclusion criterion for SSD. We, therefore, emphasize the need to investigate the association between clinical diagnoses and differences within criteria with or without health anxiety. 

We found that being exposed to rape was associated with a higher risk of having FSD than being exposed to groping, although the CIs overlap. There seems to be a clear trend toward a higher risk of having FSD the more severe the sexual assault. This trend has also been shown in previous studies [19,32,68]. Additionally, the sensitivity analysis showed a trend towards more severe outcomes when measuring the cocktail effect of both groping and rape, with rape being the leading factor. 

The SA prevalence of 3% seen in this cohort is lower than the estimated prevalence from WHO [4]. A Norwegian prevalence study of SA [69] found that 6.7% had experienced SA. Here most victims were below the age of 30, and SA was rare above the age of 60. Furthermore, it has been documented that victims of SA are more often found in low-income families, with poorer jobs, and lower education levels [70]. As the total population in our study is at the higher end of the social status measure, we may not have been able to estimate a precise measure of SA for the total Danish population. 

Dividing the sample into females and males did not alter our main results. However, there were only 26 male victims in this study, and it can, therefore, not be ruled out that outcomes for female and male victims may differ. We, therefore, stress the need for investigating female and male victims separately.

Our results are in line with all the biopsychosocial theories listed in the introduction, e.g., deficits in emotion regulation, repression, sensitization, and the influence of toxic stress within the epigenome system [38,39]. However, our results cannot determine if any of the theories are better supported than the others or whether any of them should be discarded.

We found a high risk for having emotional distress and health anxiety after being exposed to SA; however, we found a dampening effect for health anxiety and emotional distress in the crude interaction analysis for single-organ FSD and SA as well as for CF and SA, while only emotional distress had a dampening effect on multi-organ FSD. No interaction with any other FSD was found, indicating that being exposed to SA may induce different body and mind reactions as found by Ulirsh et al. according to distinctions between PTSD and the development of somatic symptoms [6]. As there are more cases with single-organ FSD and CF, this may explain why we found interactions in these cases, while not in others, which stresses the need for larger studies to investigate these interactions. 

We found that the association between FSD and SA still existed when adjusting for physical and emotional abuse. This indicates that the association between SA and FSD is unique and is not solely the impact of a cumulative effect of traumatic events. However, by adjusting for adverse childhood environment in the main analyses, we found that we considered the complexity of the association without removing too much of the effect as adverse childhood environment and physical and emotional abuse are somewhat entwined [14,17,44,71]. Nevertheless, we stress the need for further studies with more power as there are often, as in this study, a large amount of SA victims who have also experienced physical and emotional abuse. Therefore, larger studies including more cases with experiences of sexual assault solely and no physical and emotional abuse are needed.

Adverse childhood environment may be associated with sexual assault and FSD. Thus, the risk of sexual victimization among women is associated with factors such as unemployment, homelessness, addictions to drugs or alcohol, and a low socioeconomic status [70]. Associations between socioeconomic status such as education and employment [46] and traumatic experiences are associated with FSD as well. Analyzing adverse childhood environment was introduced to adjust for social inequality in health. It can be argued that our variable is not sufficient as there are other adverse experiences related to childhood conditions than the ones we have included in the measure. However, no studies have reported associations between an increased risk of sexual victimization if one had, for instance, lost a parent or experienced a natural disaster. It would have been ideal if factors such as parental mental illness had been included; however, we did not have information on this issue. Nevertheless, the model benefits from including the complexity of childhood conditions for adverse experiences.

### 4.1. Strengths and Limitations

The main strength of this study is that it fills a gap in the field of SA and FSD. This study examined largely uninvestigated associations and may bring a new perspective on how we understand the consequences of sexual assault. In applying the combination of exploring a new area, the variety and range in our outcome measures (both symptom-specific BDS and medical specialty-specific FSS), a priori based assessment on confounding variables, a sensitivity analysis, and an interaction analysis, we have a strong theory-based adjustment for factors influencing our estimates.

Another major strength is the large random unselected sample of the general adult population comprising both sexes with an age range of 50 years. Although the participation rate was rather low in DanFunD-2 [41], the response rate of our exposure variable, i.e., the questions on SA, was very high.

The biggest limitation of this study is the cross-sectional design, which does not allow us to determine the causality. However, as the CLAM is a validated retrospective measure, we still have some time perspective, indicating a causal direction from SA (measured lifelong) to FSD (measured within the last 12 months). 

As our data on SA include the age at the time of exposure, it would have been beneficial if we could have examined data on SA by age group. However, a meta-analysis by Paras et al. found no significant age difference between exposure in childhood and adulthood for FSD, while other findings suggest that there is no difference in childhood/adolescence either [5,32]. Furthermore, it would also have been relevant to investigate the cumulative effect of SA on FSD; however, our data are pooled with both periods and single events, and it would, therefore, not be possible to investigate the cumulative effect as this would also measure a time period as a single event. Furthermore, the results would thus be weakened as the prevalence of SA victims in this cohort is too small to perform subanalyses. However, we urge future studies to investigate different effects of SA according to age at exposure, time delay, cumulative effects, and severity. 

We chose the combined SCL-8 measurement on emotional distress for our study instead of using specific measures for anxiety and depression as the primary focus of the study was to investigate FSD and secondarily to compare estimates for FSD with estimates of general emotional distress to investigate the consequences of SA in a wide biopsychosocial perspective. Furthermore, we believe it would weaken our results if this was not investigated. The interaction analysis showed wide CIs for all cases, which troubled the adjusted analysis due to over-fitting. 

Another limitation of this study is the difficulty in measuring all the complexity surrounding sexual assaults, as we did not have data on, e.g., ethnicity, gender, and sexual orientation, as it has been shown that minorities have a higher risk of experiencing SA as well as a higher risk of experiencing poor mental and physical health following the assault [72,73], mediated by the feelings of minority stress [72]. To the best of our knowledge, associations between minority stress and FSD in sexual minorities have not yet been investigated. However, we have included important measures of other trauma as confounding variables in our analysis. Furthermore, we have included a measure of the severity of SA. Another limitation might be the low overall response rate and potential selection bias as there were 88 non-participants in the sample from the CLAM [44]. Another study on the DanFunD data performed a non-responder analysis investigating mental disorders [74]. This non-responder analysis showed a tendency for non-participants to have more mental disorders (a measure different from emotional distress which was included in our study). However, mental disorders were associated with cohabitation status and non-participants were not different from participants. Therefore, there is no indication that the DanFunD data were biased by non-participants [74]. However, as individuals who were not born in Denmark or were not Danish citizens were excluded from study participation, this may have resulted in a bias towards underestimating the results, as these groups are shown to be at increased risk for SA [2]. Furthermore, the exclusion criteria may be controversial as they include different ethnicities. However, they were required, as participants needed to be fluent in Danish to read and understand the questionnaires. Furthermore, Danish citizenship and being born in Denmark were required in order to merge the data with other Danish registries and obtain sociodemographic and health data since birth, which was pivotal for the aims of the DanFunD study. Therefore, the epidemiological quality of data weighed more than the diversity of ethnicity and citizenship in the area. Still, as this study is based on a large randomly selected population cohort, apart from non-Danish citizens, the study reflects the diversity in the Danish general adult population.

All measures in our study are self-reported. Therefore, it is possible that some of our cases may represent a false positive diagnosis or conventionally defined somatic disease. Petersen et al. found that there is a higher percentage of self-reported diagnoses compared with clinical diagnostic interviews in the population; however, the study also showed that self-reported data are still suitable for screenings and paraclinical research [75]. Therefore, we consider the results to be valid although not representative of actual clinical diagnoses. 

Another potential limitation regarding the self-reported data is the possibility of recall bias. However, being subjected to SA is a very traumatic experience, and a victim may, therefore, be able to remember the incident for a long time after the assault. On the other hand, being a victim of SA can be a taboo subject, and some victims may not fill out the questionnaire truthfully, although a study discovered a tendency for SA victims to participate in research projects; yet they may be the more resourceful with the capacity to seek help (61). Therefore, there may be an information bias that may underestimate the effect. On the other hand, the rise in the prevalence of SA in Denmark according to the Department of Justice (1) and the Danish Institute of Public Health (2) can be a symptom of a de-tabooing of victimization, as there has also been an increased focus in society on de-victimization of victims of SA in recent years according to global trends such as #MeToo.

### 4.2. Implications

Regarding clinical implications, we wish to stress that although FSD is not only explained by psychological conditions, some individuals with FSD who are also victims of SA may not benefit from general treatment before coming to terms with their traumatic experience, e.g., via psychological interventions. In the psychiatric field, it has been found that traumas such as SA [76] may not be disclosed during treatment and are sometimes mistaken for other psychiatric illnesses such as schizophrenia. This often leads to an aggravation of the illness and unsuccessful treatment [76,77]. Therefore, we underline the need for more awareness of the possibility of SA as a specific risk factor in FSD, as it may interfere with successful treatment in this field as well as in others.

## 5. Conclusions

In conclusion, we found significant results for all outcomes except for CWP. Our results suggest a large impact of sexual assault on the overall somatic and psychological health of victims. Therefore, we stress the need for more focus and research in this field. In particular, prospective studies would be beneficial for investigating the causality of the association.

## Figures and Tables

**Figure 1 ijerph-20-06947-f001:**
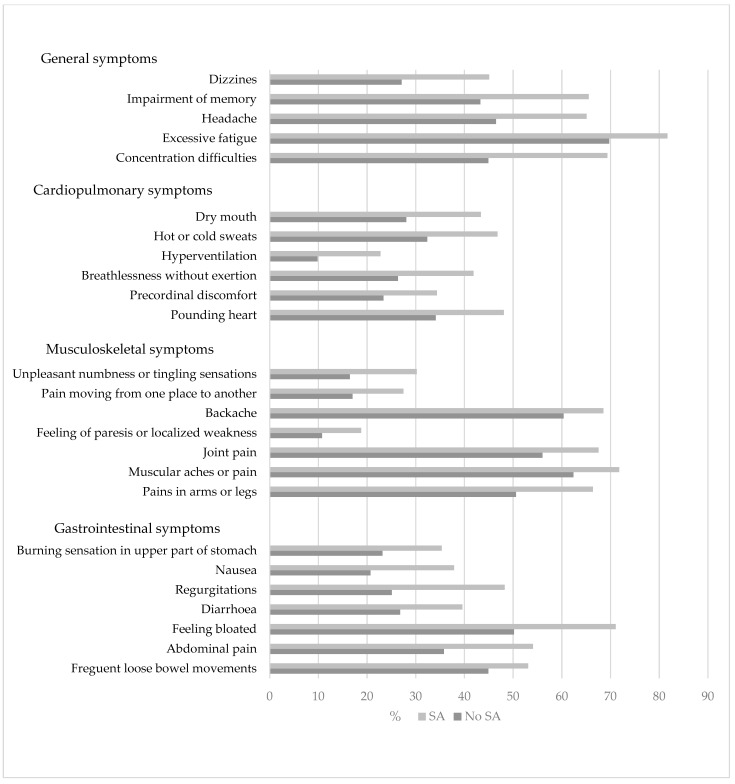
Symptom load for SA cases compared with the total population.

**Table 1 ijerph-20-06947-t001:** Population Characteristics.

Variables (Measure)	Categories	Type of Assault			
		Total Population *n* = 7493	Exposed to Sexual Assault *n* = 235	Subdivisions on Sexual Assault	
				Rape *n* = 139	Groping *n* = 179
Age at inclusion: Median (IQR) Age at exposure to SA% (*n*)	Baseline	54 (44–63)	51 (42–61)	53 (44–62)	50 (42–60)
	<5 years	--	11.6 (27)	6.5 (9)	15.1 (27)
	6–11 years	--	30.0 (70)	20.9 (29)	34.1 (61)
	12–18 years	--	39.9 (93)	38.1 (53)	38.0 (68)
	>18 years	--	33.0 (77)	34.5 (48)	12.9 (23)
Sex% (*n*)	Male	46.1 (3456)	11.1 (26)	9.35 (13)	13.41 (24)
	Female	53.9 (4037)	88.9 (209)	90.64 (126)	86.59 (155)
Vocational training% (*n*)	No education	11.1 (829)	12.8 (30)	13.0 (18)	12.9 (23)
	<3 years	15.2 (1142)	17.0 (40)	18.7 (26)	18.4 (33)
	3–4 years	42.7 (3202)	38.7 (91)	37.4 (52)	38.6 (69)
	>4 years	28.3 (2118)	27.7 (65)	27.3 (38)	26.3 (47)
	Missing 2.7 (202)				
Adverse childhood environment% (*n*)	Missing 1.7 (125)	6.6 (491)	10.6 (25)	10.8 (15)	12.9 (23)
Physical abuse% (*n*)	Missing 1.5 (115)	13.4 (1000)	50.6 (119)	60.4 (84)	53.6 (96)
Emotional abuse% (*n*)	Missing 4.1 (305)	5.9 (444)	35.3 (83)	43.2 (60)	35.8 (64)
Abuse (physical and/or emotional)% (*n*)	Missing 4.1 (305)	16.0 (1.199)	60.4 (142)	68.3 (95)	64.2 (115)
Social status Median (IQR)	Missing 1.1 (86)	7 (6–8)	6 (5–7)	6 (5–7)	6 (5–7)
Emotional distress% (*n*)	Missing 1.8 (136)	4.8 (359)	11.1 (26)	13.0 (18)	12.3 (22)
Health anxiety% (*n*)	Missing 0.5 (39)	5.6 (421)	11.5 (27)	13.7 (19)	12.3 (22)
FSD% (*n*)	Single-organ	15.8 (1.184)	6.0 (71)	4.0 (47)	4.3 (51)
	Multi-organ	1.1 (84)	15.5 (13)	10.7 (9)	14.3 (12)
	Missing 1.8 (135)				
FSS% (*n*)	CWP	4.6 (341)	8.5 (20)	10.1 (14)	9.5 (17)
	Missing 1.5 (109)				
	IB	3.6 (269)	9.4 (22)	11.5 (16)	8.4 (15)
	Missing 2.3 (171)				
	CF	9.2 (690)	22.6 (53)	27.3 (38)	25.1 (45)
	Missing 1.2 (93)				
	MCS	2.1 (156)	7.2 (17)	8.6 (12)	6.7 (12)
	Missing 1.7 (130)				
	WAD	1.6 (121)	4.7 (11)	5.0 (7)	4.5 (8)
	Missing 2.1 (160)				

SA: Sexual assault; FSD: functional somatic disorder; FSS: functional somatic syndrome; CWP: chronic widespread pain; IB: irritable bowel; CF: chronic fatigue; MCS: multiple chemical sensitivity; WAD: whiplash-associated disorder; and IQR: interquartile range. Individuals missing answers on SA items: *n* = 293.

**Table 2 ijerph-20-06947-t002:** Associations between FSD and SA.

FSD Cases	Crude			Adjusted *		
	RR	95% CI	*p*-Value	RR	95% CI	*p*-Value
Single-organ Crude *n* = 7096 Adjusted *n* = 6861	2.05	(1.68–2.51)	<0.001	1.51	(1.22–1.87)	<0.001
Multi-organ Crude *n* = 6035 Adjusted *n* = 5840	6.95	(3.91–12.32)	<0.001	3.51	(1.89–6.49)	<0.001
**FSS Cases**						
CWP Crude *n* = 7112 Adjusted *n* = 6886	1.93	1.25–2.98	0.003	1.28	0.83–1.98	0.261
IB Crude *n* = 7061 Adjusted *n* = 6829	2.76	1.82–4.18	<0.001	2.00	1.30–3.07	0.002
CF Crude *n* = 7135 Adjusted *n* = 6897	2.54	1.98–3.26	<0.001	1.81	1.42–2.32	<0.001
MCS Crude *n* = 7107 Adjusted *n* = 6872	3.77	2.32–6.14	<0.001	3.04	1.79–5.17	<0.001
WAD Crude *n* = 7075 Adjusted *n* = 6841	3.12	1.70–5.57	<0.001	2.62	1.37–5.03	0.004

* Adjusted for age at inclusion, sex, social status, vocational training, and adverse childhood environment. Abbreviations: RR: risk ratio; SA: sexual assault; FSD: functional somatic disorder; FSS: functional somatic syndrome; CWP: chronic widespread pain; IB: irritable bowel; CF: chronic fatigue; MCS: multiple chemical sensitivity; and WAD: whiplash-associated disorder.

**Table 3 ijerph-20-06947-t003:** Associations between FSD and sexual assault are divided into rape and groping.

FSD Cases	Crude						Adjusted *					
	Groping			Rape			Groping			Rape		
	RR	95% CI	*p*-Value	RR	95% CI	*p*-Value	RR	95% CI	*p*-Value	RR	95% CI	*p*-Value
Single-organ	1.94	(1.54–2.46)	<0.001	2.31	(1.83–2.92)	<0.001	1.39	(1.08–1.80)	0.011	1.64	(1.28–2.09)	<0.001
Multi-organ	8.13	(4.51–14.63)	<0.001	8.41	(4.33–16.33)	<0.001	3.76	(1.99–7.09)	<0.001	5.18	(2.73–9.86)	<0.001
**FSS Cases**												
CWP	2.15	(1.35–3.42)	0.001	2.29	(1.38–3.80)	0.001	1.35	(0.85–2.14)	0.211	1.35	(0.79–2.30)	0.271
IB	2.41	(1.46–3.98)	0.001	3.42	(2.13–5.13)	<0.001	1.68	(1.00–2.82)	0.049	2.59	(1.61–4.16)	<0.001
CF	2.82	(2.17–3.68)	<0.001	3.06	(2.31–4.05)	<0.001	1.89	(1.46–2.45)	<0.001	2.14	(1.64–2.79)	<0.001
MCS	3.40	(1.92–6.01)	<0.001	4.78	(2.54–7.89)	<0.001	2.50	(1.33–4.71)	0.005	3.61	(1.99–6.58)	<0.001
WAD	2.91	(1.44–5.88)	0.003	3.32	(1.58–7.00)	0.002	2.29	(1.07–4.90)	0.033	2.97	(1.39–6.33)	0.005

* Adjusted for age at inclusion, sex, social status, vocational training, and adverse childhood environment. Abbreviations: RR: risk ratio; FSD: functional somatic disorder; FSS: functional somatic syndrome; CWP: chronic widespread pain; IB: irritable bowel; CF: chronic fatigue; MCS: multiple chemical sensitivity; and WAD: whiplash-associated disorder.

**Table 4 ijerph-20-06947-t004:** Associations between FSD and SA with abuse as a confounding factor.

FSD Cases	Sexual Assault *			Groping *			Rape *		
	RR	95% CI	*p*-Value	RR	95% CI	*p*-Value	RR	95% CI	*p*-Value
Single-organ	1.23	(1.08–1.41)	0.002	1.12	(0.98–1.29)	0.107	1.29	(1.08–1.53)	0.004
Multi-organ	2.45	(1.25–3.43)	0.009	2.54	(1.26–5.12)	0.009	3.73	(1.85–7.52)	<0.0001
**FSS Cases**									
CWP	1.05	(0.67–1.65)	0.828	1.09	(0.67–1.77)	0.736	1.07	(0.62–1.86)	0.804
IB	1.63	(1.02–2.62)	0.041	1.26	(0.71–2.25)	0.431	2.32	(1.41–3.83)	0.001
CF	1.68	(1.36–2.07)	<0.001	1.70	(1.37–2.10)	<0.0001	1.97	(1.59–2.45)	<0.0001
MCS	2.52	(1.26–4.01)	0.006	1.92	(0.99–3.72)	0.053	2.45	(1.26–4.76)	0.008
WAD	1.98	(1.00–3.91)	0.049	1.67	(0.76–3.67)	0.205	2.08	(0.95–4.59)	0.068

* Adjusted for age at inclusion, sex, other abuses (physical and emotional), social status, vocational training, and adverse childhood environment. Abbreviations: RR: risk ratio; SA: sexual assault; FSD: functional somatic disorder; FSS: functional somatic syndrome; CWP: chronic widespread pain; IB: irritable bowel; CF: chronic fatigue; MCS: multiple chemical sensitivity; and WAD: whiplash-associated disorder.

## Data Availability

Data are not available online as the research is still ongoing at the moment.

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
