# Peer review of "Are Sexual Assaults Related to Functional Somatic Disorders? A Cross-Sectional Study"

_ijerph, 2023, doi:10.3390/ijerph20206947_

Round 1

Reviewer 1 Report

Ijerph-2578613 review

Thank you for the opportunity to review this manuscript.  This is an interesting and important area of study.  Few studies have investigated the association between sexual assault and chronic pain conditions and symptoms; even fewer have made a distinction between different forms of assault.  Therefore, the study adds important information to the existing literature.  I have the following comments/concerns:

11.       The abstract states that “the incidence of sexual assault is increasing”.  Perhaps reported rates are increasing, but its unclear if the actual incidence is. 

22.      The second paragraph of the introduction is quite long and feels a bit disjointed.  Can it be organized differently or split into separate paragraphs?

33.       It would be helpful to also present the percentages of individuals who experienced only physically forced sex, only physical forced sexual contact, and both.  Were these conceptualized as separate constructs?  Or, for individuals who experienced both, were they assigned to the more serious experience (rape)?  It might be useful to run some sensitivity analyses exploring these questions – possibly including only the “single event type” groups, maybe splitting into three groups (one or both), or assigning those in the both group to the more serious subtype.

44.       Relatedly, do you have frequency information?  Typically those who experience more serious forms of assault also experience more cumulative experiences.  So, those who experienced forced rape may have had more sexual assault experiences which may be confounding the results.

55.       Can the authors better clarify the different uses of FSD (what is included) and FSS?  FSS seems clearer because it has the 5 conditions; however, FSD is unclear to me.

66.      Can the authors also clarify how the questions to assess FSD and FSS and somatic symptoms are different?  Supplemental material may be helpful to delineate these measures and constructs.  How much overlap is there between constructs?

77.       Was sex referring to sex at birth?  Or was gender assessed?  Did the survey include a question to determine individuals identifying as transgender or nonbinary?

88.       Were higher scores on subjective social status better or worse?  Including how to interpret some of these measures would be helpful.

99.     Related to the above, was the final scoring for adverse childhood experiences 0/1 (so any event would place them in a ‘yes’ category)?  Or was this summed?

110.   It seems odd to combine childhood experiences of abuse with adult experiences of abuse.  However, I acknowledge that this is simply controlling for these experiences, which is critically important to do.  How was this scored?  Is it the sum of binary variables, or did any event place them in a ‘yes’ category.  I would have more concerns if any abuse event (childhood or adulthood) simply put them in a ‘yes’ category – if this is the case, I would consider splitting up the events.

111.   Was the impact of sexual assault on outcomes different for men vs. women?

112.   The incidence of 3% for sexual assault seems quite low.  The criteria of physical force or threat of force likely resulted in lower rates.  But, were the authors surprised by this?  Can the authors place this in context with other population estimates?

113.   In the descriptive analyses, the findings related to emotional distress/health anxiety are reported in the opposite direction.  Can the authors instead state how many individuals who experienced SA had high emotional distress or high anxiety?

114.   Figure 1 seems to have white blocks that are obscuring some of the information.

115.   Lines 282-285, the authors provide separate information on somatic symptoms for those who experienced groping vs. rape – could this be provided as supplemental material?  Again, may benefit from creating 3 groups or dealing with overlap in some way.

116.   I see that the authors chose not to include time at incident.  However, did the authors consider that the stronger association is possibly due to the severity of the event overlapping with recency?  May be important to discuss.

117.   Additionally, do the authors have any other information regarding sexual assault experiences that may help readers understand the complexity of these measures?

Minor comments:

118.   Make sure all the condition acronyms are defined the first time they are mentioned.

119.   Please review the manuscript for small typos.  For example, “was” on line 201 should be “were”; Line 347 it says “seen shown”; Line 450, “FSD is not psychologically explained conditions…” – this should not be plural. 

The manuscript should be reviewed for small typos as indicated in my review. 

Reviewer 2 Report

The authors present important information regarding the health impact of sexual assault on victims.  This information is important and relevant to mental health treatment and healthcare providers.  Since there are (so) many abbreviations in this manuscript, I would suggest creating a listing of them (as an appendix) so that the reader can keep track.

Reviewer 3 Report

The authors have done good work on a topic in need of research. Functional somatic disorders are common in both general medicine and mental health services and we have few tools to treat them. Furthermore, its relationship with trauma, although assumed, is insufficiently validated empirically.

My biggest reservation about the article is that the possible influence of sex is insufficiently discussed. Both sexual assault and functional somatic disorders are more common in women. The authors themselves state that approximately 9 out of 10 study subjects who have suffered sexual assault are women. When sex is introduced together with other variables in the regressions, the RR decreases, although the effect of each of the adjustment variables has not been studied separately. If this is not possible, good practice from a gender perspective requires that, at a minimum, results be presented disaggregated by sex. If it is not possible with the regressions due to the low sample, at least with the descriptive ones, even if it is in supplementary materials. And the possible influence of sex would have to be commented on in the discussion.

Minor issues that could improve the understanding of the article are:

-The numbering of the sections at the end of the introduction: line 92 should not have it and therefore the following must be changed.

-Exclusion criteria are not justified. The first two can be controversial.

-Lines 127-130: If FSD is used to refer to BDS and FSS, it does not make sense that it is also used to refer to only BDS. Perhaps it is a typo and at the beginning of line 129 the F should be changed to B. If not, please review and specify what each term is used for.

-Line 141 appears to require a separate heading.

-Lines 141-149: Please specify for each disorder whether it is evaluated with a validated questionnaire or the criteria are applied directly.

-I don't quite understand why the age of exposure to sexual assault is not included in the analyses. How are the case sizes different from the rest of the variables? The authors justify that there are usually no differences between exposure in childhood and adolescence. But are there differences between childhood/adolescence and adulthood?

-Table 1: I am struck by the high number of missing values in the Adverse Childhood Environment variable. Although to a lesser extent, the variables Emotional Distress, BDS, CWP, IB, CF, MCS and WAD also present many missing variables.

-Figure 1: Some of the titles are not visible or are cut off.

-I would change the order of the results, putting all the descriptive results first and ending with the regressions.

-Lines 282-285: Results are not shown even though they have implications for the second secondary hypothesis.

-I find it difficult to understand section 3.5. My question is: Are the same variables that have been controlled in the previous regressions (age, sex, etc.) controlled or not?

-Lines 329 and 331: I think CFS is CF.

-Lines 347-349: This explanation is not clear. Please specify which group or groups these subjects are included in.

-Line 422: There is talk of mental disorders. Were they assessed and not included in the analyses, or does it refer to the overall measure of psychological stress?

-Lines 449-450: This phrase needs to be qualified as it is controversial given that there is no empirically demonstrated etiology.

I hope these suggestions help you. It has been very interesting to read your work.
